# Human-Derived Cortical Neurospheroids Coupled to Passive, High-Density and 3D MEAs: A Valid Platform for Functional Tests

**DOI:** 10.3390/bioengineering10040449

**Published:** 2023-04-06

**Authors:** Lorenzo Muzzi, Donatella Di Lisa, Matteo Falappa, Sara Pepe, Alessandro Maccione, Laura Pastorino, Sergio Martinoia, Monica Frega

**Affiliations:** 1Department of Informatics, Bioengineering, Robotics, and Systems Engineering (DIBRIS), University of Genoa, 16145 Genoa, Italy; lorenzo.muzzi@dibris.unige.it (L.M.); sergio.martinoia@unige.it (S.M.); 23Brain AG, 8808 Pfäffikon, Switzerland; 3Corticale Srl., 16145 Genoa, Italy; 4Department of Experimental Medicine (DIMES), University of Genoa, 16132 Genoa, Italy; 5Department of Clinical Neurophysiology, University of Twente, 7522 NB Enschede, The Netherlands; 6Department of Human Genetics, Radboudumc, Donders Institute for Brain, Cognition, and Behaviour, 6500 HB Nijmegen, The Netherlands

**Keywords:** microelectrode arrays, electrophysiology, 3D neuronal network, neurospheroids, brain-on-a-chip, rapid differentiation, h-iPSC, functional test

## Abstract

With the advent of human-induced pluripotent stem cells (hiPSCs) and differentiation protocols, methods to create in-vitro human-derived neuronal networks have been proposed. Although monolayer cultures represent a valid model, adding three-dimensionality (3D) would make them more representative of an in-vivo environment. Thus, human-derived 3D structures are becoming increasingly used for in-vitro disease modeling. Achieving control over the final cell composition and investigating the exhibited electrophysiological activity is still a challenge. Thence, methodologies to create 3D structures with controlled cellular density and composition and platforms capable of measuring and characterizing the functional aspects of these samples are needed. Here, we propose a method to rapidly generate neurospheroids of human origin with control over cell composition that can be used for functional investigations. We show a characterization of the electrophysiological activity exhibited by the neurospheroids by using micro-electrode arrays (MEAs) with different types (i.e., passive, C-MOS, and 3D) and number of electrodes. Neurospheroids grown in free culture and transferred on MEAs exhibited functional activity that can be chemically and electrically modulated. Our results indicate that this model holds great potential for an in-depth study of signal transmission to drug screening and disease modeling and offers a platform for in-vitro functional testing.

## 1. Introduction

The human brain fascinates with its complexity, but the limited accessibility represents one of the major issues in the comprehension of its mechanisms. Therefore, studies of neuronal electrophysiological activity on simplified and more controlled in-vitro models are a fundamental step towards understanding the functioning of brain tissue. Microelectrode arrays (MEAs) technology is a well-established tool to study how cellular composition, connectivity, genetic, and epigenetic expression correlate with the functional electrical activity expressed by in-vitro neuronal models [1,2,3,4,5,6,7,8]. MEAs applicability ranges from drug/toxicological screening [1,5,9] to the characterization of various neuronal disorders [3,8,10,11,12,13,14,15,16,17,18,19,20,21,22,23,24,25,26,27,28,29,30,31,32,33,34,35,36]. In particular, with the introduction of human-induced pluripotent stem cells (hiPSCs) and differentiation protocols, human-derived neuronal models could be created, potentially making the in-vitro approach more reliable and representative of in-vivo conditions. This allows one to generate and study neuronal cells carrying the precise genetic information of the donor in a non-invasive way, easing translation of results to the clinic and limiting the use of animal experimentation [37,38]. Most of the studies on neuronal models have been performed on monolayer cultures. Although these models are widely accepted and provide invaluable results, they are inherently unable to replicate the three-dimension (3D) environmental complexity of the brain (i.e., cell morphology, extra-cellular matrix, and axons-dendrites’ extension in the 3D space) and yet the in-vitro results are not always congruent with those in-vivo [39,40].

Different methodologies to build 3D models have been sought with the aim to investigate neuronal functions in a more in-vivo-like condition [41,42,43]. Most of the strategies for 3D tissue construction are scaffold-based (i.e., ECM- [44,45,46,47,48,49,50,51] or microbeads-based [52,53,54,55,56]) in which the used materials allow the spontaneous formation of 3D networks with arborizations in the 3D space. Human iPSCs are also able to self-assemble without the use of 3D scaffolds into brain organoids, which are 3D spheroidal aggregates composed by the heterogeneous population of cells with a cytoarchitecture resembling the embryonic human brain [57,58,59,60,61]. Different methodologies are adopted to generate brain organoids, from the use of spinner vessel bioreactors [62,63,64,65], round bottom wells and/or non-adherent [66,67,68,69,70], micro-molding and/or microfluidics, cell aggregates on adherent substrates [71,72,73], and hanging drop [74,75,76]. Furthermore, several guided methods have been proposed to induce hiPSCs to differentiate with a certain regional specificity (i.e., cerebral cortex, hippocampus, and midbrain) [57,59,60,77,78]. In the last years, several groups have seen the advantage of using brain organoids to establish 3D in-vitro models of brain disease [79,80,81].

MEAs devices have been used to characterize the electrophysiological activity of 3D neuronal networks in healthy conditions [54,55,56,64,82,83,84]. The activity exhibited by brain organoids have been characterized also in models of neurological diseases, such as genetic syndromes [85], psychiatric disorders [86,87], and neuro-degenerative diseases [88,89]. Different types of MEAs devices have been used with 3D models (i.e., with passive or active, planar or 3D, and low- or high-density electrodes), but yet it is not trivial to choose the one providing the most valuable and accurate information about the exhibited activity.

The electrophysiological dynamics of neuronal networks result from the interplay between different cellular populations. In neurological diseases, specific cell-types are affected, leading to neuronal network phenotypes [90,91,92,93,94]. Three-dimensional models, allowing one to obtain new insights into the cell-type specific contribution to neuronal network dynamics in healthy and diseased conditions, are needed to improve our understanding of the pathophysiological phenotype and underlying molecular pathways and to develop targeted treatment strategies [95,96,97,98,99]. In the past few years, several protocols have been developed to force the differentiation of hiPSCs into a homogeneous cellular population (i.e., glutamatergic [100,101], GABAergic [33,102,103], dopaminergic [104,105] motor neurons [106], and astrocytes [107,108]). This allows one to choose the cell types which are most relevant to the physiological mechanisms or the pathophysiology of the disease under investigation and combine them at will. To date, studies on the cell-type specific contribution to neuronal network phenotypes have been performed using monolayer cultures [8,33,34,35,36,109,110]. In addition, we proposed a scaffold-based method to generate 3D neuronal networks in which the cellular composition and density are controllable and that can be coupled to MEAs for chronical electrophysiological recordings [55].

Here, by adapting the scaffold-based protocol published by us [55], we propose a scaffold-free methodology, allowing the generation of 3D neuronal networks from hiPSCs with control over the cellular composition and suitable for chronical electrophysiological recordings. We demonstrate that hiPSCs-derived neurons and rat astrocytes are able to aggregate into neurospheroids. We show that neurospheroids exhibit spontaneous activity that can be modulated by receptor blockers or excited by electrical stimulation. In addition, we compared the electrophysiological activity recorded by three MEAs with different electrode characteristics (i.e., passive or active, planar or 3D, and low- or high-density). We found that high-density MEAs (HD-MEAs) provide a more accurate analysis of the networks’ dynamics, allowing for single-unit investigations, the tracking of action potentials, and the estimation of connectivity maps.

## 2. Material and Methods

### 2.1. Human-Induced Pluripotent Stem Cells Generation and Maintenance

We used a characterized rtTA/Ngn2-positive hiPSC line generated from the fibroblast of a healthy subject (30 year-old female) that was kindly provided in frozen vials by Frega et al. [101]. This line was reprogrammed via episomal reprogramming (Coriell Institute for medical research, GM25256). Afterwards, rtTA/Ngn2-positive lentiviral vectors were used to stably integrate the transgenes into the genome of the hiPSCs. Maintenance was performed as previously described [101].

### 2.2. Neuronal Differentiation and Neurospheroids Generation

Neurospheroids were composed by 800,000 early-stage excitatory cortical layer 2/3 neurons and astrocytes in a 1:1 ratio (Figure 1). Early-stage neurons were obtained by inducing the overexpression of the neuronal determinant Neurogenin 2 (*Ngn2*) in hiPSCs upon doxycycline treatment for three days and were co-cultured with astrocytes, as previously described [55]. We exploited the hanging-drop method to generate spherical aggregates of cells, and we used a 5 cm petri dish as a ‘moisture chamber’ for the neurospheroids. We half-filled the petri dish with DPBS and we used the inner part of the lid to sustain the drops of a medium used as a scaffold-free culture. In detail, we inverted the lid, and we placed eight 15 µL drops of the neurobasal medium into the inner part of the lid. Thereafter, we added 15 µL of the mixed-cells solution (i.e., solution of 5,330,000 cell/mL composed of 500,000 early neurons and 500,000 astrocytes) to each drop previously placed. Consequently, we placed the lid back on the petri dish with a gentle but rapid movement to avoid sliding of the drops. We then stored the dish into the incubator at 37 °C and 5.5 CO_2_. We considered this day as Day in Vitro (DIV 0) of the neurospheroids. On DIV 1, we added Ara-C (Sigma-Aldrich, St. Luois, MO, USA) to each drop to a final concentration of 2 µM. During the first week in vitro, we added the medium if evaporation occurred. From DIV 7 on, we replaced 10 µL of the medium for each drop, paying attention to avoid damaging the forming neurospheroids. At DIV 10, neurospheroids were moved into 24 well-plates pre-coated with a 1% *w*/*v* Alginate solution to avoid cell attachment and preserve the spheroidal shape. From this moment on, the medium was supplemented with 2.5% of FBS, and 30% of the medium was refreshed three times a week. At DIV 35, neurospheroids were moved into MEAs.

### 2.3. MEAs Devices

Neurospheroids were plated to MEA60s (Multi Channel System, MCS, GmbH, Hamburg, Germany), 3D-MEA60s (Multi Channel System, MCS, GmbH), and Accura High Density MEAs (HD-MEAs) from 3Brain AG, Switzerland. MEA60s are glass devices where 60 titanium nitride electrodes (diameter of 30 µm and inter-electrode distance of 200 µm) are embedded in the center of the culture well. They are arranged in an 8 × 8 grid, resulting in an active area of 1.6 mm × 1.6 mm. The 3D-MEA60s have 60 pyramidal electrodes (height of 100 µm, diameter tip of 12 µm, and inter-electrode distance of 250 µm) disposed as in the MEA-60s. HD-MEA (Accura model, 3Brain AG, Pfäffikon, Switzerland) devices present an active area of 3.8 mm × 3.8 mm in which 4096 CMOS electrodes (dimension of 21 µm × 21 µm and pitch of 60 µm) are integrated and arranged in a 64 × 64 grid.

Before plating, the MEA60s and 3D-MEA60s were sterilized in the oven at 120 °C for 2 h, while the HD-MEAs were sterilized with 70% EtOH for 20 min. The active areas of the devices were coated with poly-L-ornithine (100 µL drop of a 50 µg/mL solution, 4 °C overnight) (Sigma-Aldrich) and human-laminin (80 µL drop of a 20 µg/mL solution, 4 °C overnight) (BioConnect, Toronto, ON, Canada).

### 2.4. Data Acquisition and Analysis

Spontaneous activity was recorded at DIV 42–49–56 for 10 min, while modulated activity was recorded at DIV 57–60 (Figure 1C). Recordings were performed using the 2100 System (MEA 2100-System, MCS) for the MEA60s and 3D-MEA60s and the BioCam Duplex (3Brain AG) for the HD-MEAs. Data were sampled at 20 kHz and filtered at 200 Hz using a 2nd order Butterworth filter. Incubator-like conditions were maintained during recording by keeping the culture at 37 °C and 5.5% CO_2_.

*Spike detection.* For recordings obtained by the MEA60s and 3D-MEA-60s, spike detection was performed using the Precise Timing Spike Detection (PTSD) [111]. BrainWave software (v.4.4, 3Brain AG) was used to compute the PTSD on the data acquired from the HD-MEAs. A peak-lifetime period of 1 ms, a refractory period of 1 ms, and a standard deviation factor of 6 and 8 for passive MEAs and HD-MEAs, respectively, were used as detection settings. The mean firing rate (MFR) was evaluated for each electrode as the ratio between all the detected spikes and the recording time. We considered the analysis of only the ‘active’ electrodes exhibiting a MFR > 0.1 spike/s.

*Burst detection*. We considered a burst as a series of three consecutive spikes firing no more than 50 ms apart from each other. The mean burst rate (MBR) was evaluated as the sum of all the detected bursts that occurred in an active channel, divided by the recording time, and we considered bursting the electrodes of those having a MBR > 0.2 burst/min. The mean burst duration (MBD) was computed by averaging all the burst durations detected in the whole culture. The percentage of random spikes (PRS) was obtained as the ratio between the total number of non-burst spikes and the total number of spikes and the percentage of bursting channels (PBC) as the ratio between bursting electrodes and active electrodes.

*Network burst detection*. Synchronous events occurring within the network, defined as Network bursts (NBs), were detected as previously reported in [55].

*Post-Stimulus Time Histogram*. To evaluate the electrical response, the Post-Stimulus Time Histogram (PSTH) was computed by counting all the detected spikes occurring after each stimulation (i.e., bin of 10 ms). The fast and late responses were computed by counting the evocated spikes occurring in the first 100 ms and between 100 ms and 1000 ms, respectively, while the peak latency was obtained as the time of the maximum activation. The PSTH was normalized by evaluating the mean value of the pre-stimulus time histograms (i.e., 200 ms before each stimulation, bin of 10 ms). Each normalized PSTH was then averaged across the 50 stimulations delivered and then across neurospheroids cultured on the same MEAs.

*Cross-correlation*. To evaluate the functional connections between electrodes, we implemented a cross-correlation (CC) script in Matlab using a common normalization factor [112]. Connectivity matrices were obtained by analyzing the correlograms in a time window of 300 ms centered at time 0 and using a 0.2 ms bin. In order to eliminate the contribution of spurious connections, a threshold (i.e., average of the weights of the total connections plus 2 times the value of the standard deviation) was applied. For each matrix the total number of links, their average weight, and the average connection time τ (i.e., time lag) were calculated. To reduce the computational time, we applied the CC algorithm to 144 electrodes arranged in a 12 × 12 square grid and placed under the neurospheroids.

*Single-cell signal propagation*. After spike sorting, based on PCA and k-means clustering, individual units (i.e., neurons) with an average peak amplitude of at least 300 µV were taken into account. For each considered unit, the spike timestamp was computed, and the raw traces of the entire electrode array in a time window of 10 ms around each spike were extracted. All the extracted raw data traces were then averaged. The traveled distance of each propagation was calculated as the sum of the geometric distances between consecutive electrodes involved in the propagation. The average speed was evaluated by dividing the time difference between the first and last spike involved in the propagation by the traveled distance.

### 2.5. Electrical Stimulation and Neuromodulation

At DIV 57, functional tests on the neurospheroids were performed (Figure 1C). The effect of three neuromodulators—50 µM of 6-cyano-7-nitroquinoxaline-2,3-dione (CNQX, Sigma-Aldrich), 80 µM of D-2-Amino-5-phosphonopentanoic Acid (D-APV, Sigma-Aldrich), 5 µM of Kainic Acid (KA, Sigma-Aldrich)—was evaluated both in terms of spontaneous activity and in response to electrical stimulation. The protocol started with 10 min of recording in spontaneous conditions, followed by a stimulation protocol (i.e., 50 bipolar current pulses of a 60 µA peak-to-peak amplitude, 0.1 Hz, half-width of 100 µs, delivered from two electrodes under the neurospheroids). Consequently, the cultures were treated with one neuromodulator and placed in the incubator for 30 min. After this time, 10 min of spontaneous activity in the presence of the neuromodulator was recorded, followed by the current stimulation protocol. Finally, three washing steps with the conditioned medium were performed to remove the neuromodulator.

### 2.6. Immunofluorescence

After the electrophysiological recordings, the samples were fixed directly on the HD-MEAs at DIV 60, and fluorescence images were acquired. For the fixation, the samples were exposed to a 4% paraformaldehyde solution (PFA, Sigma-Aldrich) for 20 min at room temperature and then washed three times in a phosphate-buffered saline (PBS, Sigma-Aldrich) solution. The samples were permeabilized with 0.2% triton X-100 (Thermo Fisher Scientific, Waltham, MA, USA) for 15 min. The samples were exposed to a Blocking Buffer Solution (BBS, composed of 0.5% fetal bovine serum (Sigma-Aldrich) and 0.3% bovine serum albumin (Sigma-Aldrich) in PBS) for 45 min at room temperature. We used GFAP (diluted 1:500, Sigma Aldrich) and MAP-2 (diluted 1:500, Chemicon Millipore, Temecula, CA, USA) as primary antibodies to mark glial and neuronal cells, respectively, and Dapi (diluted 1:10,000, Sigma) to label nuclei. We used Alexa Fluor 488 (diluted 1:700, Thermo Fisher Scientific) and Alexa Fluor 546 (diluted 1:1000, Invitrogen, Waltham, MA, USA) and Goat anti mouse or Goat anti rabbit as secondary antibodies.

### 2.7. Statistical Analysis

In this work we show data obtained from n = 12 neurospheroids generated from the same batch. Six samples were grown in the free culture while the other 6 were grown on MEA60s, 3D-MEAs, and HD-MEAs. A statistical analysis was performed on data obtained from all active electrodes. After the evaluation of the normality test was carried out with GraphPad Prism, we performed a non-parametrical Wilcoxon signed-rank test. Differences were considered significant when *p* < 0.05.

## 3. Results

### 3.1. Human iPSCs-Derived Neurons Aggregate into Neurospheroids

First, we monitored the formation of the neurospheroids over time, and we observed that hiPSCs-derived neurons started aggregating after about 10 days of culturing in the petri dish. When moved, into free-cultures, the aggregates that showed a spherical shape with a mean diameter of 412 ± 16 µm and 429 ± 20 µm at DIV 40 and DIV 62, respectively (Figure 2A,B). Then, we investigated the development of the neurospheroids on MEAs. The neurospheroids adhered to all MEAs, and we observed that the height of the structure was lower than the mean diameter measured from the samples kept in the free culture (Figure 2C). In particular, we found that the highest point of the neurospheroids growing on a HD-MEAs is at 180 µm from the electrode plane and that the diameter was ~550 µm (Figure 2C,D). Furthermore, we observed that both neurons and astrocytes were present within the neurospheroids and that cells were viable in the inner part since nuclei were round and regular through the entire height of the structure (Figure 2C,D and Appendix A).

### 3.2. Neurospheroids Showed Electrophysiological Activity on MEAs

The neurospheroids were plated on the three different MEAs (Figure 3A–C) at DIV 35 to compare the functional activity recorded by the three devices. We observed that neurospheroids exhibited electrophysiological activity that could be detected by all devices at DIV 42. By comparing the raw data of a single electrode under the neurospheroids, we observed different levels of noise and spike amplitude. In particular, HD-MEAs detect significantly larger waveforms as compared to the lower resolution devices. This fact, combined with the higher resolution of the HD-MEAs, allows one to clearly identify the onset of network events (i.e., darker vertical bands in raster plots, Figure 3G–I).

We then followed the development of the neurospheroids by recording the spontaneous activity at different time points (DIV 42–49–56). We found that the number of active electrodes and most of the firing metrics tend to increase over time in neurospheroids plated on HD-MEAs, while they remain almost stable in the MEA60 (Figure 4A–D). In all cases, we observed that activity evolved over time, moving from mainly random spiking to bursting. Indeed, the percentage of random spikes (PRS) decreased over time while the percentage of active bursting channels increased (Figure 4G–H). With all devices, we were able to detect phenomena of synchronous network activity, although with the MEA60 and the 3D-MEA60, they only emerged later in development (i.e., from DIV 49) and with a significantly lower frequency as compared to HD-MEAs (Figure 4E,I–K). Even if the duration of the detected NBs was variable between the different devices (Figure 4F), the average shapes of the NBs were similar but are very scattered in the case of the MEA60 and the 3D-MEA60 (Figure 4L).

### 3.3. Neuromodulation Affects Spontaneous Activity and Electrical Induced Activity

Once the neurospheroids reached a steady-state regime (i.e., presence of network bursts 3D [8,34,55]) at DIV 57, we subjected them to an electrical stimulation conjugated with chemical modulation protocols, in order to evaluate the effects of three neuromodulators on functional activity (Figure 5 and Appendix A). The inhibition of the AMPA receptors (i.e., CNQX) led to a significant reduction of the mean firing rate and bursting rate (Figure 5C,D). In addition, the synchronous network activity was suppressed (Figure 5F,G), causing an increase in random activity and a significant decrease in active and bursting electrodes (Figure 5A,H,I). As regards the effect on electrical stimulation, the response to stimuli in the presence of CNQX is slower as compared to the basal condition (Figure 5J, peak latency) with a significant reduction of evoked potentials in the first 100 ms following the application of the stimulus. The suppression of NMDA receptors (i.e., APV) did not affect the number of active electrodes and did not significantly reduce the mean firing rate and mean bursting rate (Figure 5C,D). The synchronous activity also remained almost unchanged while the random activity generally decreased (Figure 5F,H). By evaluating the change in response to electrical stimulation, we found a significant delay in reaching the maximum response peak and a decrease in spikes evoked in the first 100 ms (Figure 5K). Finally, Kainic acid had a significant effect on the synchronous network activity, increasing its rate while maintaining a similar average duration of events (Figure 5F,G). There was also a substantial decrease in active electrodes. Evaluating the response to the stimulus, the results showed an effect similar to that caused by CNQX, with an anticipation of the peak and a reduction in rapid evoked potentials, but it did not show significant differences regarding the late response phase (Figure 5K).

### 3.4. Neuromodulation Affects Connectivity and Signal Propagation

To evaluate how neuromodulation and electrical stimulation affected the connectivity of the neurospheroids, we performed a cross-correlation analysis. Results indicate that the inhibition of AMPA receptors with CNQX led to a decrease in the number of links and an increase in the mean weight (Figure 6B,C), while the electrical stimulation led to an increase of the number of links and a slight decrease in the mean weight (Figure 6E,F). A comparison between the electrical stimulation in a spontaneous condition and the CNQX stimulation led to a decrease in the total number of links and an increase in their mean weight (Figure 6H,I). The blockage of NMDA receptors with APV caused a less significative change of connectivity in the absence of stimulation. The application of the electrical stimulation led to a decrease of the number of links, an increase of their mean weight, and a slight reduction of the mean lag when compared to the electrical stimulation in the basal condition. Electrical stimulation under the APV effect resulted in decreasing the number of links, increasing their mean weight, and slightly decreasing their mean lag compared to electrical stimulation in the basal condition (Figure 6H–J). Finally, electrical stimulation after KA administration led to a decrease in the number of links and a slight increase in their mean weight.

Thanks to the high spatial resolution offered by the HD-MEAs, it was possible to highlight, trace, and estimate the speed of action potentials in the various experimental phases (Figure 6L–N). Under spontaneous conditions, we found propagations covering mean lengths of 375 ± 194 µm with mean speeds of 0.44 ± 0.30 m/s (Figure 6M,N). In the presence of CNQX, we calculated higher speeds with lower variability (0.7 ± 0.15 m/s) but over wider mean distances (636 ± 362 µm). In the presence of APV, we found shorter propagations (278 ± 112 µm), with an average velocity similar to that estimated with CNQX (0.65 ± 0.29 m/s). Finally, with KA, we found average propagation lengths of 510 ± 300 µm with velocities comparable to spontaneous condition (0.39 ± 0.16 m/s).

## 4. Discussion

Here, we present a scaffold-free method to create 3D neuronal networks from hiPSCs in a rapid manner that allows the manipulation of cellular composition and is appropriate for chronic electrophysiological recordings through MEAs.

The interplay between different cellular populations is responsible for the electrophysiological dynamics displayed by neuronal networks. In particular, in neurological diseases, specific cell-types are affected, leading to neuronal network phenotypes [90,91,92,93,94]. In the past few years, different methodologies to build 3D human models have been developed to investigate the neuronal network dynamic in healthy and diseased conditions. Spheroidal aggregates composed by a heterogeneous population of cells (i.e., brain organoids [57,58,59,60,61,62,63,64,65,66,67,68,69,70,71,72,73,74,75,76,77,78,79,80,81]) have been used, taking advantage of the capability of stem cells to self-aggregate in a scaffold-free environment. However, brain organoids showed several limitations, including high variability due to a limited control over the final size, cellular composition (i.e., cell type and ratio), and density of neural aggregates [113], extremely long protocols (i.e., up to several months) [64,82], and a lack of vascularization, affecting the diffusion of oxygen and nutrients and resulting in a necrotic core [61,114]. To gain a deeper understanding of the pathophysiological phenotype and the underlying molecular pathways, and to create targeted treatment strategies, it is important to study the cell-type specific contribution to neuronal network activity in a controlled manner, which is not possible with the protocols for spheroidal aggregates currently available. Protocols to differentiate hiPSCs specific cell-types have been developed, allowing one to generate 2D neuronal networks in which cellular composition and ratio can be fully controlled and engineered at will [33,34,55,100,101].

In the spheroidal aggregates proposed in this work, neurons were generated from hiPSCs through *Ngn2* induction and were co-cultured with astrocytes. This is a rapid, reproducible, and efficient protocol, allowing one to obtain functionally mature neuronal networks composed by a homogeneous population of cortical excitatory neurons and astrocytes with a controlled density in a short time (i.e., 5 weeks [101]). Other differentiation protocols generating other homogeneous cellular populations can be used in combination with the one based on *Ngn2* induction [33,101,103], offering a broad spectrum of designability for 3D tissues. Functional neurospheroids were obtained by adapting the scaffold-based protocol for the rapid generation of functionally engineered 3D human neuronal network as published by us [55]. Whereas in the scaffold-based protocol, early-stage neurons and astrocytes were plated directly on MEAs together with micro-beads, in the protocol presented here, we used the hanging-drop method to avoid the use of a scaffold and to be able to move the structure freely into any substrate. With the scaffold-based approach, we showed that the ratio between neurons and astrocytes remained stable over development, that neurons were interconnected to each other forming a network [55], and that they exhibited a complex electrophysiological dynamic when coupled to MEAs [55]. Here, we showed that neurons and astrocytes formed spheroid aggregates exhibiting activity comparable with the one of 3D neuronal networks built with the scaffold-based method [55], indicating that the network formation is similar.

The use of the hanging-drop technique made it possible to generate the neurospheroids without the use of bioreactors or scaffolds. This methodology allows one to obtain many uniform spheroidal aggregates that can be cultured in parallel in the same plate, thereby reducing the variability and allowing high-throughput screenings [66]. The cells needed 7 to 10 days to form a spheroidal structure inside the drop. After 10 days, all the drops in the petri dishes contained solid and regular structures and were moved to non-adherent multi-well plates. This was mainly due to the difficult maintenance of the neurospheroids in the hanging-drop condition. In particular, cells that had a small volume of the medium available are easily subjected to evaporation, and mechanical stresses could damage the viability of the 3D network. Although we did not find significant size differences between the generated samples, we did notice a slight increase in size over time (see Figure 2B). This might be due to a proliferating astrocytic part, which recovered after the initial phases of aggregation of the neurospheroids where we treated it with Ara-C. The shapes of the nuclei in the core of the neurospheroids were round and regular (Figure 2D, Appendix A), indicating that cells were healthy. This is an essential point in neurospheroids’ generation in which the core represents the most sensitive part, because it is subjected to necrosis due to the lack of nutrient exchange [113]. We also noticed that the samples on MEA devices adapted their dimensions and preferred a radial growth on the substrate rather than preserving the spherical shape. In fact, after 30 days in culture, we found a maximum height of the neurospheroids of 180 µm with a diameter of about 550 µm, suggesting a flattening of the structure on the electrode plane, that was also confirmed by the increased number of active electrodes in the MEAs out of the spheroid area.

In the past few years, different MEA devices have been used to record the electrophysiological activity of 3D neuronal networks [45,64,70,73,82,83,86,87,88,89,115,116,117,118]. Passive MEAs allowed one to obtain recordings only from a few scattered electrodes [45,64,70,73,82,83,86,87,88,89], while HD-MEAs provided a more accurate estimation of the neuronal network dynamics, allowing single-unit analysis, the tracking of action potentials, and the estimation of connectivity maps [115,116,117,118,119,120]. However, the electrophysiological activity was recorded from sliced organoids instead of intact structure maps [115,116,117,118]. Even if this approach allows one to access the inner part of the tissue, it is debatable if organoid slice preparations contain sufficient preservation of the anatomical organization necessary to reproduce the physiologic metrics of intact organoids. Furthermore, the variability in results might increase because of the comparison between different slices, and the experimental design possibilities are limited since the developmental trajectory of a single sample cannot be followed for a long time.

Here, we verified the functionality of the neurospheroids, and we monitored the development of spontaneous electrophysiological activity for 5 weeks (i.e., from the moment of adhesion to the device) on MEAs. This allowed us to investigate whether coupling these structures with MEAs could constitute a valid 3D in-vitro model useful for functional characterizing. In addition, to highlight which type of MEAs was the most suitable for studying their electrophysiological dynamics, we used three MEA devices with different electrode characteristics. To reduce variations due to different neuronal and astrocytic preparations [8], we compared the activity of neurospheroids belonging to the same batch. We used two types of MEAs with passive electrodes, one with planar (MEA60) and one with 3D electrodes penetrating the structure and recording signals internally (3D-MEA60). In addition, to achieve a high-density spatial resolution and a higher signal to noise ratio, we used a device exploiting the CMOS technology (HD-MEAs). We moved the culture to the devices at DIV 35 because, at this stage of development, a steady-state and stable network dynamic had been observed in 2D and 3D [34,55]. During the first few days after adhesion, we were unable to record activity. This was probably due to the fact that cells must adhere to the electrode plane to develop synapses and functional connections before a signal can be picked up. After a week, we found active electrodes in all the devices and already a quite mature network dynamic, presenting single channel activity and burst. Over time, the active bursting channels increased and PRS decreased, showing trends in line with the results of other 3D cultures in the literature [54,55,56]. During the recording weeks, several electrodes became active, even far away from the neurospheroids. Many cells moved from the neurospheroids to the electrode plane, creating a peripheral 2D network from which activity can be detected (see Figure 2). We observed differences in levels of the firing rate recorded from the three MEA devices. This derives from the different nature and density of the electrodes. In particular, the raw signal recorded by MEA60s appears noisier as compared to HD-MEAs, making it difficult to detect close spikes in moments of high synchronous activity such as NBs. Consequently, lower electrode density combined with a lower signal/noise ratio led to a different quantification of the MBR, MBD, and NBR in the passive MEAs as compared to HD-MEAs. Similar considerations apply to 3D-MEAs, with the difference that the active electrodes are located inside the neurospheroids and can, therefore, record activity by cells in all directions. Thus, our results showed that a small number of electrodes failed to fully describe the dynamics of these cultures, such as defining a correct MFR or the onset of NB during development. The choice of the device is crucial to obtain correct information about the electrophysiological dynamics of 3D structures, in particular when comparing healthy and diseased conditions or the effect of compounds on the observed phenotype. Thus, HD-MEAs should be preferred since they provide the most valuable and accurate information about the exhibited activity. Instead, the use of devices with lower electrode densities might lead to a not complete or incorrect characterization of the neuronal network dynamics.

Once we verified that the neurospheroids were active and mature in electrophysiological terms, both internally and externally, we evaluated their response to electrical stimuli in conjunction with neuromodulators. Although we have performed these experiments on all the samples in each type of MEA, we showed, in the main text, only the data obtained from the HD-MEAs. Indeed, the HD-MEAs resulted in the most suitable device to record electrophysiological activity, thanks to the higher spatial resolution allowing a more accurate analysis. Interestingly, neurospheroids on 3D-MEA60s detached after the application of the first stimulation protocol. This might be caused by variations in the electrode impedance due to the production process and the fine tip, which can alter stimulation intensities, especially with currents. It is, therefore, possible that the actual stimulation current was higher than the set value, causing irremediable damage to the neurospheroids.

Since neurospheroids are composed of only excitatory neurons and astrocytes, we used neuromodulators that interact with glutamatergic receptors. CNQX and APV, respectively, inhibit AMPA and NMDA receptors, which we know are the main mediators of the network activity of these cultures [34]. The administration of CNQX led to a radical change in the network dynamics of the neurospheroids. The spontaneous AMPA-inhibited activity showed a significant decrease in network properties compared to the baseline, almost completely suppressing single-channel and network bursting activity. The inhibition of NMDA receptors alone did not significantly affect the spontaneous activity of the neurospheroids. The effect of the modulator is observed in a decrease of about 40% in the percentage of random spikes and a reduction in the average length of the NBs. This reflects the results of previous studies [70,82], showing that NMDA and AMPA receptors are involved in the generation of single-channel/network bursting and that their balance must be maintained to preserve the network dynamics [34,121,122,123].

Previous studies regarding the electrical response of dissociated cultured network show that two phases of the electrical response can be distinguished [4,124,125,126]. The first phase (i.e., early response) occurs within the first 80 ms and is related to the response of the subpopulation on neurons that are directly excited from the electrical stimulation. The late phase occurs after the 80 ms and was found to be mediated mainly from inhibitory connections that propagate the stimulation of the entire network [4,124,125,126]. In our model, in which GABAergic neurons are not present, we have found that the response to the stimulus was mainly mediated by AMPA receptors. In fact, AMPA receptors drive the fast dynamics, while on the contrary, the NMDA receptors drive the slower dynamics [123,126,127]. Under the effect of CNQX the neurospheroids modified the response to the electrical stimulus, not only by decreasing the amount of evocated spikes, but also by changing their response shape. The maximum response peak shifted below 20 ms, because only neurons in close contact with the stimulating electrodes were elicited, and they were unable to communicate quickly with others, resulting in a failure in mediating the response to the whole network. With APV treatment, the response to the stimulus changes: fewer evoked spikes in the first 100 ms and a delay in reaching the maximum peak were observed. The inhibition of spontaneous fast activity allows for a more precise estimation of propagation speeds and the identification of longer trajectories. This might be cause by the removal of the background noise coming from excitatory postsynaptic potentials, and, therefore, the average of the raw data centered on a single unit sorted into a channel was less influenced by the activity of other neurons in the surroundings [119,120]. Kainic acid is known to cause convulsions in in-vivo rats [128], and it is utilized to model epileptic-like events in-vitro (i.e., intense initial bursts followed by repetitive after-discharges) [9,20,122,129,130,131,132]. The general effect of KA in the activity exhibited by the neurospheroids was a significant lowering of the number of active electrodes and a consistent increase in the NBR. Furthermore, the response to the stimulus appeared flatter when compared to that in baseline conditions, and the response peak is squashed in the first 20 ms. These two aspects are in line with previous findings on 2D human-derived neuronal networks [131,132]. The connectivity analysis revealed that the electrical stimulation during KA strengthened the connections by decreasing the number of links, increasing their weight, and decreasing their lag. These results suggest that epileptic-like events can be successfully induced in our neurospheroids with KA, but further studies are needed to confirm that this is a valid model for studying KA-induced seizures. In fact, all previous works [9,20,122,129,130,132,133] have used KA in cultures where GABAergic neurons were present, while we are describing the results derived from a purely excitatory 3D culture.

The study of the connectivity in-vitro is known to be essential to understand the basic mechanism of memory and learning [134,135,136,137]. The high resolution of HD-MEAs allowed us to evaluate the connectivity maps of neurospheroids in-vitro. The inhibition of AMPA receptors increased the disorganization of the network by increasing the number of connections and decreasing their weight. We observed that inhibiting fast transmission greatly decreases the number of connections, and only those with heavier weights are preserved. The inhibition of NMDA receptors also led to a decrease in the number of links, with an increase in the weight of the connections but in a less clear way. We also recorded a decrease in the average lag of these connections, a clear sign that most connections under spontaneous conditions are fast connections mediated mainly by AMPA. The stimulation in the presence of APV instead led to a consolidation of the connections as the number of links decreased, preserving only the connections with the highest weight. This gives a further confirmation of how AMPA and NMDA receptors are also involved in the processes of plasticity and memory [127,138,139,140,141] and makes our model usable also for this type of study [125,137,142,143].

The proposed methodology needs to be intended as a proof of concept and presents some limitations. First, neurospheroids were composed only by a homogeneous population of excitatory neurons in co-culture with astrocytes. The integration of other protocols [33,101,103] allowing the forced differentiation into different neuronal populations was not presented. Another limitation lies in the difficulty of performing recordings in a complex 3D space. In this work, the electrophysiological dynamics exhibited in the inner part of the neurospheroids was recorded only in a single plane as all the micro-electrodes have the same height (i.e., 100 µm). New technologies allowing recordings of the electrophysiological activity of 3D neuronal cultures from multiple 2D planes are emerging [45]. However, devices providing a high level of spatial and temporal resolution necessary to fully harness the potential of 3D models are lacking. Finally, a low sample size has been used in this work. However, neurospheroids were generated following scaffold-based protocol previously presented by us [55], using a robust and reproducible method to create neuronal networks with low variability and similar patterns of activity on MEAs [8,101]. The scaffold-based and scaffold-free methods showed comparable results (i.e., activity), indicating that similar neuronal networks were formed.

## 5. Conclusions

We have shown a proof of concept for generating neurospheroids that can be potentially engineered at will in a rapid manner, without the use of scaffolds or bioreactors, and that can be coupled intact with MEAs devices for functional analysis. Confocal images show a lack of necrotic tissue and the electrophysiological characterization of the spontaneous and modulated/stimulated activity indicated that functional 3D neuronal networks were formed. We highlighted the importance of using HD-MEAs to better assess the network dynamics and to extrapolate more complex information. The proposed model will allow the investigation of the interplay between different cellular populations and their contribution into the neuronal network dynamics in healthy and diseased conditions in 3D in a short time, which is promising for deepening our understanding into neuronal network phenotypes and underlying molecular pathways, and for developing cell-specific targeted treatment strategies.

## Figures and Tables

**Figure 1 bioengineering-10-00449-f001:**
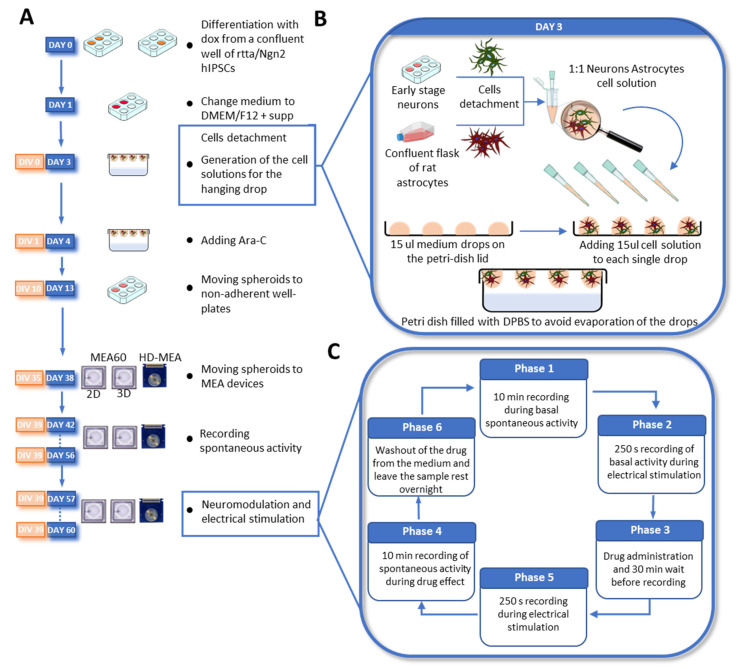
Overview of the differentiation steps and experimental design. (**A**) Timeline of the development of the neurospheroids from the differentiation process to the transfer and growth onto the MEAs. (**B**) Main steps performed at DAY 3/DIV 0: Collection of cells and hanging-drop technique. Early-stage neurons and astrocytes are harvested the same day in two separate tubes. Consequently, cells are counted and mixed in a 1:1 ratio in a 1 mL vial at a final concentration of 5,330,000 cell/mL. Therefore, 15 µL of the cell suspension were plated on 15 µL medium drops in the lid of a petri dish. Finally, the lid was gently placed back on the petri dish, half-filled with DPBS, and then stored in the incubator. (**C**) Experimental protocol for chemical neuromodulation and electrical stimulation.

**Figure 2 bioengineering-10-00449-f002:**
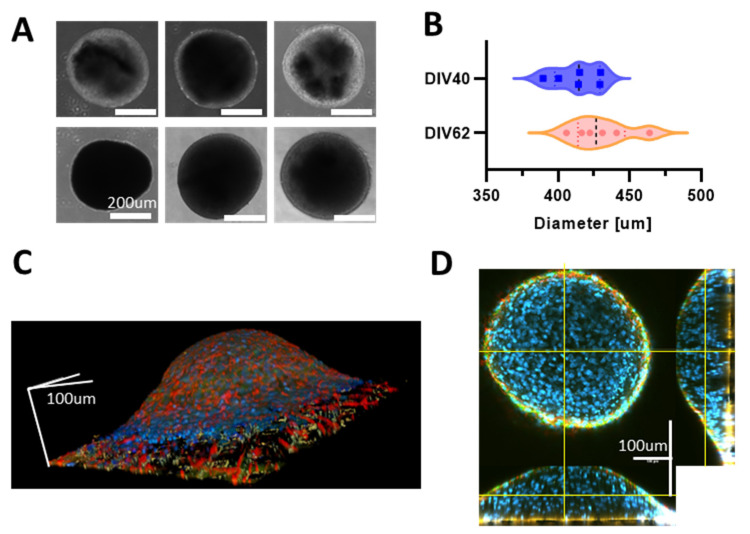
(**A**) Images of different neurospheroids in the free culture during development. The first row shows pictures obtained at DIV 40 and second row at DIV 62. Scale bar, 200 µm. (**B**) Evaluation of the mean diameter of n = 6 neurospheroids at DIV 40 and DIV 62 in the free culture. (**C**) Reconstruction of the 3D volume of a neurospheroid plated on a HD-MEAs at DIV 35 and fixed at DIV 62. Images were acquired with a confocal microscope using a z-stack of scale bars that represent 100 µm in each direction. (**D**) Orthogonal view of the sample shown in (**B**). Yellow lines represent the observation plane. In blue: DAPI, green: MAP2, red: GFAP. Scale bars represent 100 µm in each direction.

**Figure 3 bioengineering-10-00449-f003:**
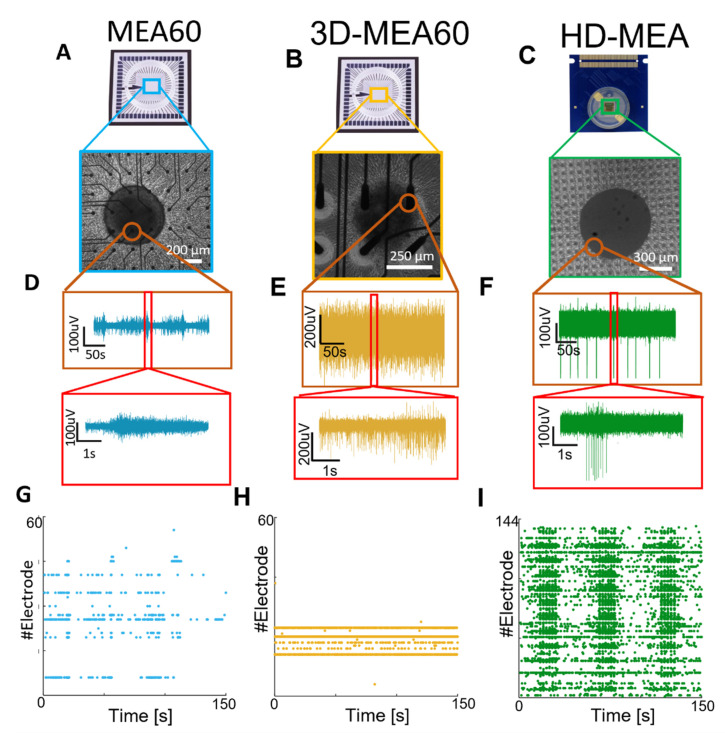
Qualitative comparison between the activity recorded by the MEA60, the 3D-MEA60, and HD-MEAs. (**A**–**C**) Picture showing a neurospheroid at DIV 49 on (**A**) the MEA60 (scale bar 200 µm), (**B**) the 3D-MEA60 (scale bar 200 µm), and (**C**) HD-MEAs (scale bar 300 µm). (**D**–**F**) Exemplificative raw data traces recorded by one electrode placed below the neurospheroid growing on (**D**) the MEA60, (**E**) the 3D-MEA60, and (**F**) HD-MEAs. A 5 min and 5 s close-up of activity are shown in the orange and red boxes, respectively. (**G**–**I**) Raster plot showing 150 s of activity exhibited by a neurospheroid recorded by (**G**) the MEA60, (**H**) the 3D-MEA60, and (**I**) HD-MEAs. Each dot represents a detected spike at the given electrode index (y axis).

**Figure 4 bioengineering-10-00449-f004:**
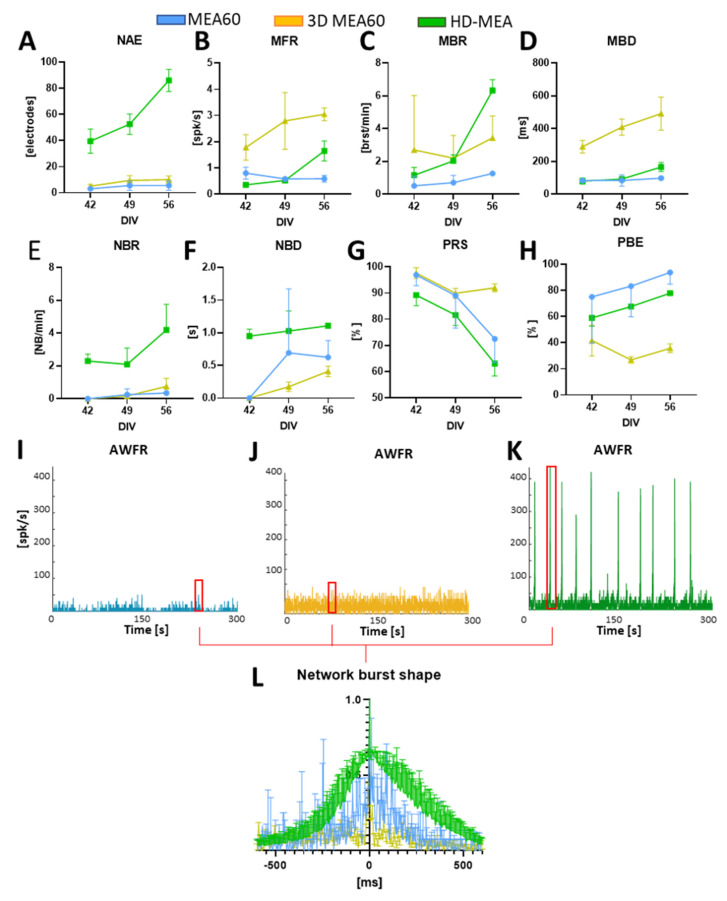
Analysis of the spontaneous activity of n = 2 samples on the MEA60 (data showed in blue), n = 2 on HD-MEAs (data showed in green), n = 2 on the 3D-MEA60 (data showed in yellow). (**A**–**H**) Graphs showing the (**A**) number of active electrodes (NAE), (**B**) mean firing rate (MFR), (**C**), mean bursting rate (MBR), (**D**) mean burst duration (MBD), (**E**) network burst rate (NBR), (**F**) network burst duration (NBD), (**G**) percentual random spikes (PRS), and (**H**) percentual bursting channels (PBC). Data are shown as a mean and standard deviation of the mean. (**I**–**K**) Graphs showing the Array Wide Firing Rate in neurospheroids growing on (**I**) the MEA60, (**J**) the 3D-MEA60, and (**K**) HD-MEAs. It quantifies the level of activity as the averaged firing rate of all the networks evaluated in 100 ms bins. Network burst (NB) are highlighted by red rectangles. (**L**) Mean NB shape aligned and normalized by the higher peak.

**Figure 5 bioengineering-10-00449-f005:**
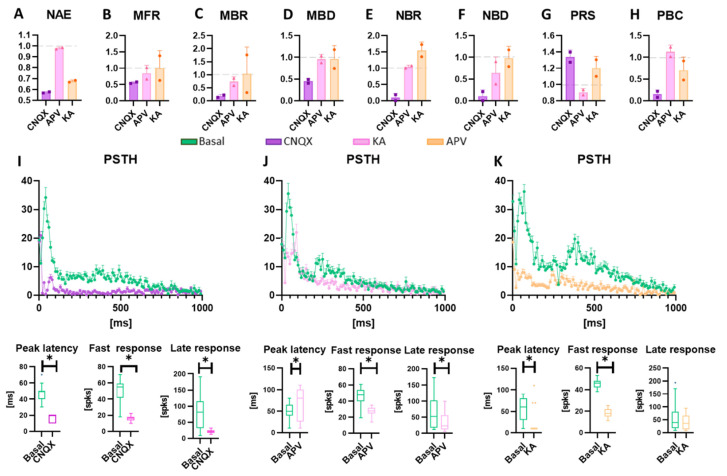
Evaluation of the electrical and chemical modulation of the activity. Data are obtained from experiments on HD-MEAs (n = 2) and are normalized to the value detected in spontaneous conditions in the absence of drugs. Data obtained with the MEA60 are reported in Appendix A. (**a**–**h**) Graphs showing the (**a**) number of active electrodes (NAE), (**b**) mean firing rate (MFR), (**c**) mean bursting rate (MBR), (**d**) mean burst duration (MBD), (**e**) network burst rate (NBR), (**f**) network burst duration (NBD), (**g**) percentual random spikes (PRS), and (**h**) percentual bursting channels (PBC) in neurospheroids treated with CNQX, APC, and KA, represented in purple, pink, and orange, respectively. Data are represented as a mean and standard deviation of the mean. (**i**–**k**) Graphs showing the effect of the modulation on the response to the electrical stimulus induced by (**i**) CNQX, (**j**) APV, and (**k**) KA. In each panel, the Post-Stimulus Time Histogram (PSTH), showing the response of neurospheroids to the stimulation, is shown (not treated and treated with CNQX, APV, and KA are represented in green, purple, pink, and orange, respectively). The peak latency represents the mean latency to reach the maximum peak in the PSTHs, the fast response box plot represents the number of evocated spikes in the first 100 ms of the PSTH, while the late response box represents the number of evocated spikes between 100 ms and 1000 ms. * *p* < 0.05.

**Figure 6 bioengineering-10-00449-f006:**
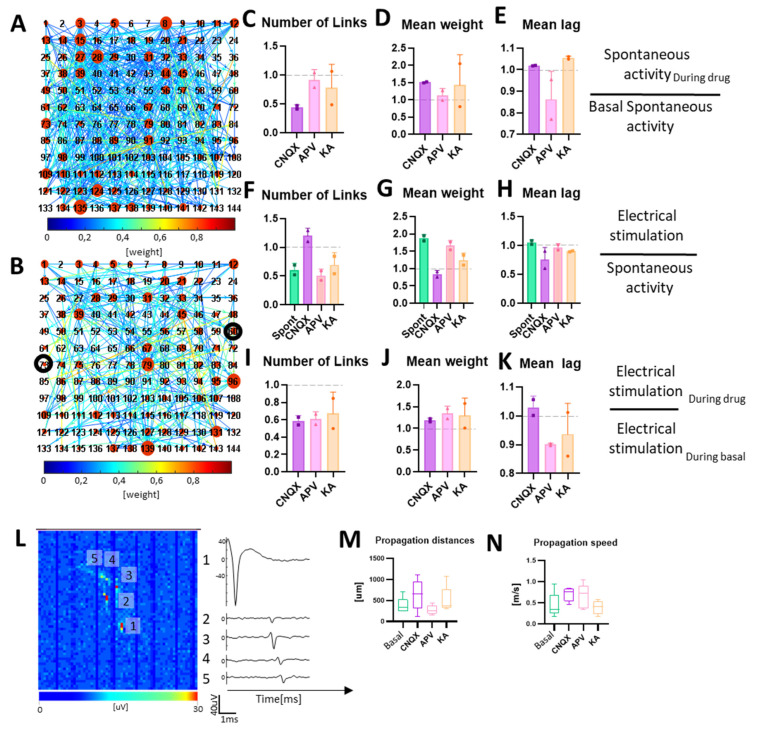
Neuronal connectivity in neurospheroids. (**A**,**B**) Connectivity matrix obtained from a neurospheroid grown on HD-MEAs at DIV 57 during (**A**) spontaneous basal activity and (**B**) electrical stimulation. Each red circle represents an electrode; the larger the size of the red circle the greater the number of links. Colormap represents the weight of each link. The darker circle indicates electrodes used for delivering stimulation. We restricted the connectivity analysis to 144 electrodes arranged in a 12 × 12 square grid, placed under the neurospheroids. (**C**–**E**) Graphs showing the effect of neuromodulators on connectivity, in particular: (**C**) the number of links, (**D**) mean weight, and (**E**) mean lag, evaluated on 10 min of activity under the effects of the drugs and normalized on 10 min of spontaneous electrical activity without drug. (**F**–**H**) Graphs showing the effect of electrical stimulation during the neuromodulation on connectivity, in particular: (**F**) the number of links, (**G**) mean weight, and (**H**) mean lag, evaluated on 10 min of recording during electrical stimulation under the effects of the drugs (or no drug for electrical stimulation, labelled as basal) and normalized with data obtained from the recording of spontaneous activity in the same conditions. (**I**–**K**) Graphs show the overall effect on connectivity of the electrical stimulation during neuromodulation, in particular: (**I**) the number of links, (**J**) mean weight, and (**K**) mean lag, evaluated during the electrical stimulation under the effect of the neuromodulator, normalized by the data obtained from recording during electrical stimulation without drugs. Bars represent the mean value, dots represent single experiments, and errors are expressed using standard deviation. (**L**) On the left, the activity map of the HD-MEAs, showing the averaged axonal propagation of a single neuron. The image is composed of 64 by 64 pixels, where each pixel represents the voltage amplitude of an electrode**.** Scale bars represent maximum variation of voltage in a 1 ms sliding window. On the right, voltage traces of five electrodes, recording the signal propagation. Numbers identify the electrodes selected on the activity map. (**M**,**N**) Graphs showing the (**M**) mean propagation distance and (**N**) mean propagation speed of ten neurons in basal and chemically stimulated conditions.

## Data Availability

Data will be available on request.

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
