# Peer review of "Human-Derived Cortical Neurospheroids Coupled to Passive, High-Density and 3D MEAs: A Valid Platform for Functional Tests"

_bioengineering, 2023, doi:10.3390/bioengineering10040449_

Round 1

Reviewer 1 Report (Previous Reviewer 2)

The authors have improved the paper and I do not have any further comments. 

Reviewer 2 Report (Previous Reviewer 1)

The manuscript can now be accepted.

Reviewer 3 Report (New Reviewer)

I had a privilege to review an article »Human-derived cortical neurospheroids coupled to passive, high-density and 3D MEAs: a valid platform for functional tests«

The article presents an interesting method to generate neurospheroids of human origin using human-induced pluripotent stem cells (hiPSC). The neurospheroids present a scaffold-free 3D model of human neuronal tissue which can be further manipulated and evaluated using different types of MEAs.

The authors were able to modulate the activity of the neurospheroids and succeeded in showing its functionality. The protocol of the study is accurately presented and the protocol carefully executed. The results are interesting and promising – the presented technology can indeed be used for further studies of brain tissue functionality.

Overall, the presented article is well-written with novel information regarding the neurospheroid culturing and evaluating. I propose the article to be published in the presented form.

This manuscript is a resubmission of an earlier submission. The following is a list of the peer review reports and author responses from that submission.

Round 1

Reviewer 1 Report

1-      The introduction needs improvement with recently published literature review. The purpose of the research is not well reported in the final part of the introduction. The motivation of the paper is not sufficiently justified in the introduction.

2-      The paper needs more clarification. The contribution of the study is not clear? What is precisely proposed in the study, and what is the actual contribution to the literature? It should be explained clearly. It would be excellent if the importance of this issue was validated by detailed research and thoroughly documented data.

3-      The Limitations of the proposed study need to be discussed before conclusion.

4-      The findings in "Conclusions" Section should be stated point by point. A contextualization has to be added as incipit, in order to make the Conclusions section self-standing. Please make a few-line conclusion about the work. It can be the essence of all the results. It should be suggestive about the best practice for future works.

5-      Author should add separate section regarding future outlook and specific comment point wise based on their study.

6-      Some grammatical errors and typos should be corrected.

7-      It is necessary to check the format of the journal.

Reviewer 2 Report

The paper describes free culture of neurospheroids and their electrophysiological activity characterisation.  My comments are:

·       This is a long paper, and I recommended to reduce its length, specifically, the materials section is very long and results and discussion sections have overlaps.

·       The quality of figures 3 G, H, I, need to be improved.

·       It would be great if authors can reflect how theirs results can help accelerating the use of the discussed methods in drug discovery, and clinical diagnostics, …

Reviewer 3 Report

This paper presents a functional analysis of neurospheroids obtained from hiPSCs in three different MEAs.

Comments:

1) There are many typos and unusual notations throughout this paper.

2) The neurospheroid used is also few in number and shows no robustness in the data presented. The sample size is too small because the characteristics of hiPSC vary greatly depending on the degree of differentiation.Sample size is small and not appropriate. How was the sample size determined?

3) The authors are "the first" to present results from a major microelectrode array (MEA) and claim to have characterized the electrophysiological activity exhibited by neurospheroids, but many papers have been published in the past. The novelty is low.